# Supramolecular Structure of Polypropylene Fibers Extruded with Addition of Functionalized Reduced Graphene Oxide

**DOI:** 10.3390/polym12040910

**Published:** 2020-04-14

**Authors:** Jan Broda, Janusz Fabia, Marcin Bączek, Czesław Ślusarczyk

**Affiliations:** Institute of Textile Engineering and Polymer Materials, University of Bielsko-Biala, Willowa 2, 43-309 Bielsko-Biala, Poland; jfabia@ath.bielsko.pl (J.F.); mbaczek@ath.bielsko.pl (M.B.); cslusarczyk@ath.bielsko.pl (C.Ś.)

**Keywords:** polypropylene, reduced graphene oxide, calcium pimelate, β modification, β-nucleating agent

## Abstract

An effective β-nucleating agent for polypropylene crystallization was obtained by the functionalization of reduced graphene oxide with calcium pimelate. The nucleating ability of the modified reduced graphene oxide (rGO-CP) was confirmed during non-isothermal crystallization. In further examinations, the rGO-CP was used as an additive to modify polypropylene fibers. The fibers were extruded in laboratory conditions. Gravity spun fibers containing three different concentrations of the rGO-CP and fibers taken at three different velocities were obtained. The supramolecular structure of the fibers was examined by means of calorimetric and X-Ray Scattering methods (DSC, WAXS, and SAXS). The considerable amount of β-iPP was obtained only in the gravity spun fibers. In the fibers extruded at higher velocities, the diminishing impact of the additive on the fibers structure was revealed. The changes observed in the fiber structure in connection with the impact of the additive on polypropylene crystallization was discussed.

## 1. Introduction

For many years polypropylene fibers have been some of the most important synthetic fibers commonly produced using the classic melt spinning technique. The fibers exhibit several favorable properties, which allow for their application in the production of various technical textiles [1]. During many years, the fibers have been repeatedly modified, as a result of which some shortcomings have been eliminated and new advantageous properties have been achieved. 

The common procedure used for the modification of polypropylene fibers consists on the application of additives intentionally added to extruded melt during fiber formation [2]. Additives were of great importance since the beginning of fiber production. Adding pigments solved the problem of the lack of fiber dyeability [3]. By the application of light stabilizers the poor UV stability was corrected [4]. Then, thanks to various additives, antioxidants, antistatic agents, flame retardants, organic and inorganic fillers, and many other properties were improved [5]. In this way, the fibers were adapted to specific requirements and the scope of application expanded considerably. 

In the recent years, a new group of additives based on graphene derivatives has attracted great attention worldwide. Due to the prominent chemical, thermal, mechanical, and electrical properties, graphene derivatives are considered excellent reinforcing fillers for polypropylene composites, which can greatly improve their electric conductivity as well as mechanical and thermal properties [6,7,8]. 

The group of graphene-based additives includes graphene oxide (GO) and reduced graphene oxide (rGO). GO is usually obtained by oxidation of graphite powder. The compound maintains the layer structure of the graphite. Nevertheless, the layers are buckled and the interlayer spacing between them is about two times bigger. Graphene oxide contains several epoxide, hydroxyl, carbonyl, and carboxyl groups, which are placed on its basal plane and edges. Reduced graphene oxide is obtained through the chemical, thermal, or electrical treatment of graphene oxide. As a result of this reduction, the oxygen functional groups are partially removed. Due to the removal of oxygen, the functionality of the interlayer space is declined [9,10].

The oxygen-containing functional groups of GO and rGO prevent a restoration of the van der Waals interaction between the particular graphene layers, enhance interactions with the polar polymer matrices, and at the same time limit dispersion in the non-polar polymers. To promote the homogeneous dispersion and enhance the interfacial interaction with hydrophobic polypropylene, the functionalization of graphene oxide with organic compounds with long aliphatic chains is required. An example of such procedure is introducing linear aliphatic chains into the nucleophilic substitution reaction [11] and grafting dodecyl amine onto the surface of graphene oxide using a solution mixing method [12].

By adding alkylated graphene oxide into polypropylene, a low content of the β modification was obtained [13]. The presence of the β modification is advantageous to macroscopic toughness and has a beneficial effect on impact strength, elongation at break, and heat deformation temperature of polypropylene products [14].

Formation of β modification is also of great interest with respect to polypropylene fibers. These fibers usually present the most stable monoclinic α modification and/or less ordered mesophase. The β modification was obtained in fibers formed under a particular cooling manner and fibers colored with quinacridone pigment extruded at very low velocities [15,16,17,18].

Formation of the β modification requires specific crystallization conditions. The available literature presents some techniques of the formation of β modification [19,20]. The most common and efficient method consists of the application of selective β-nucleating agents. Compounds which possess the nucleating ability are classified into four groups: organic pigments, aromatic amide compounds, dicarboxylic acids with their salts, and rare earth compounds. 

Pristine graphene-based additives do not nucleate the β modification. Graphene derivatives belong to α-nucleating agents which, mixed with crystallizing polypropylene melt, promote quick growth of the α crystals. With the supplementation of additional selective β-nucleating agents, the competition between β-nucleation and α-nucleation on graphene-based compounds was observed [21]. The nucleation ability of graphene oxide exceeds the efficiency of conventional β-nucleating agents, and thus it was impossible to obtain a composite with a high content of the β-form in this way. 

Obtaining β modification in polypropylene composites requires preparation of selective β-nucleating agents based on functionalized GO. For this purpose, the *N*,*N*′-dicyclohexyl-1,5-diamino-2,6-naphthalenedicarboxamide, representing the second group of β-nucleating agents, was applied [22,23]. Similar to other representatives of this group, the compound does not possess reactive groups which can be used to attach compounds directly to the surface of graphene oxide. For this reason, it was attached in a two-stage process via previous functionalization with octadecylamine.

An easier procedure to obtain graphene derivatives with the nucleating ability toward β modification consists on the application of calcium pimelate, a well-known compound from the third group of β-nucleating agents. Calcium pimelate, as well as salts of other dicarboxylic acids and various metals, has been often investigated [24,25,26,27,28,29]. In first examinations, it was discovered that pimelic acid alone could not induce the formation of β modification and that adding the second component is necessary [30]. Further examinations of various dicarboxylic acids and their salts revealed that the nucleating efficiency depends on the quantity of carbon atoms in the acids. The highest nucleation ability was shown for acids containing five to eight carbon atoms [31]. In other investigations, the high nucleating efficiency was observed for compounds with a (001) spacing from a range of 11–13 Å, closely related to the distance between helices of the same hand in the β modification of polypropylene [32]. In the compounds, polar and nonpolar parts of the salts form the surface system of ditches, which accommodate the polypropylene chains easily and facilitate formation of the β nuclei.

Efficient β-nucleating agents were prepared by mechanical mixing carboxylated graphene with calcium pimelate and by mixing suspension of calcium pimelate with carboxylated graphene dispersed in ethanol [33]. In other investigations, calcium pimelate was connected with the surface of unmodified graphene oxide in reaction with pimelic acid and calcium hydroxide [34]. The β-nucleating agent obtained in this reaction effectively promoted formation of β modification even at low additive concentration. 

In our previous investigations, calcium pimelate was attached to the surface of GO and rGO [35]. During examinations, the compounds were characterized and their nucleating ability in the quiescent state by non-isothermal crystallization of isotactic polypropylene was confirmed. In further investigations, the functionalized GO was applied as an additive by formation of melt spun polypropylene fibers [36]. In the investigation presented in the present paper, the second compound rGO functionalized with calcium pimelate was used. The compound was added as an additive by formation of polypropylene fibers formed in laboratory conditions by different compound concentrations and different take-up velocities. In the paper, the supramolecular structure of the fibers affected by the addition of the functionalized reduced graphene oxide was analyzed. To our knowledge similar studies have not been conducted so far.

## 2. Materials and Methods 

### 2.1. Preparation of Functionalized Reduced Grapheme Oxide

The reduced graphene oxide functionalized with calcium pimelate was obtained from the graphite powder according to the scheme presented in Figure 1.

In the first stage, oxidation of the graphite powder was performed. The oxidation was carried out in laboratory conditions according to the modified Hummers’ method [37]. The graphite powder flakes with a particle size of <20 µm, purchased from Sigma-Aldrich, were applied. The powder was added to sulfuric acid (H_2_SO_4_). The suspension was stirred in an ice bath. Subsequently, potassium permanganate (KMnO_4_) was slowly added. Then, distilled water and a solution of hydrogen peroxide (H_2_O_2_) were added. Finally, graphene oxide (GO) was purified through washing with distilled water. 

In the second stage, thermal reduction of GO was performed. The reduction was carried out at low temperature under a nitrogen atmosphere. 

In the third stage, reduced graphene oxide (rGO) was modified with pimelic acid and calcium hydroxide. rGO was mixed with pimelic acid and then calcium hydroxide (Ca(OH)_2_) was added. The mixture was homogenized and then heated in an aluminum vessel.

More details on the reagent’s concentration and the parameters of particular treatments during oxidation, reduction, and functionalization are presented in previous papers [35,38].

### 2.2. Formation of Polypropylene Fibers

Functionalized reduced graphene oxide (rGO-CP) was blended with polypropylene resin to obtain a masterbatch containing 5 wt.% of fine dispersed compound. Blending was performed for 15 min at a temperature of 230 °C in the barrel of the laboratory extruder. 

As a polypropylene resin, commercial isotactic polypropylene Moplen HP462R (LyondellBasell) characterized by MFI = 25 g/10 min was applied. For the formation of fibers, classic melt spinning was used. The fibers were extruded by means of the laboratory twin-screw extruder EHP-2x16S (Zamak Mercator, Poland) coupled with a 32-hole spinneret with a hole diameter of φ = 0.2 mm. The fibers were extruded at a temperature of 230 °C, cooled in the gaseous atmosphere at room temperature of 20 °C, and taken to a final winding device located in a distance of 1.5 m from the spinneret. The series of as-spun and gravity spun fibers were obtained. The gravity spun fibers were extruded at a velocity of 4 m/min without additional drawing and collected below the spinneret in a distance of 0.5 m. The fibers taken to the winding device were wound at three different speeds: 50, 200, and 800 m/min.

The fibers with different concentrations of the rGO-CP, 0.1%, 0.5%, and 1%wt, were obtained.

### 2.3. Materials Characterization

#### 2.3.1. Scanning Electron Microscopy

The morphology and elemental composition of graphene-based materials were examined by Scanning Electron Microscopy (SEM) and Energy Dispersive X-ray Spectroscopy (EDS). During the investigations, the Phenom ProX scanning electron microscope coupled with the PhenomWorld EDS detector was applied. The SEM and EDS examinations were performed in the image and map mode, respectively. On the basis of EDS examinations, the atomic percentage of atoms in graphene-based materials was determined. 

#### 2.3.2. Fourier Transform Infrared Spectroscopy

Changes in the chemical composition of compounds after oxidation, reduction, and functionalizing were analyzed by means of Fourier Transform Infrared Spectroscopy (FTIR). For the examinations, the Nicolet 6700 spectrometer (Thermo Electron Corporation) equipped with photoacoustic device MTEC was used. The FTIR spectra were registered in a range from 4000 to 400 cm^−1^ at the resolution of 4 cm^−1^ with 128 scans per spectrum. For data collection and post-processing, the OMNIC software (v. 8.0, Thermo Electron Corp.) was applied.

#### 2.3.3. Differential Scanning Calorimetry

The nucleating ability of rGO-CP during the non-isothermal crystallization of polypropylene in the quiescent state and the thermal properties of the fibers were examined by means of the Differential Scanning Calorimetry (DSC). The investigations were conducted by means of an analytical system (TA Instruments) with a calorimeter (MDSC 2920) coupled with the refrigerated cooling system. The analysis of the DSC curves was performed by means of the Universal V4.5A software. During crystallization examinations, the samples were cooled after 210 °C annealing for 5 min, with a cooling rate of 5 K/min. Based on the registered curves, the temperature and enthalpy of crystallization were determined. During examining the fibers, the samples were heated from –40 °C to 210 °C at a rate of 10 K/min in nitrogen purge (flow 40 ml/min). Based on the curves, the melting temperature and the enthalpy of transitions were determined. The content of ordered phases was calculated as the ratio of the enthalpy measured to the melting enthalpy of the fully crystalline polymer. The total crystallinity was calculated as a sum of contents of two polypropylene modifications. According to the literature, Δ*H*_m_ = 207 J/g [39] for the α modification and Δ *H*_m_ = 199 J/g for the β modification were assumed [40].

#### 2.3.4. Wide-Angle and Small-Angle X-ray Scattering

The supramolecular structure of the polypropylene fibers was examined using the wide-angle (WAXS) and small-angle (SAXS) X-ray scattering methods. 

For WAXS measurements a URD-65 Seifert diffractometer was applied. The CuKα radiation generated by radiation source was monochromatized with a nickel filter and a graphite crystal monochromator. The scattered radiation was registered by a scintillation counter in the range from 3° to 40° in steps of 0.1° and the registration time of 20 s per step. The measurements were carried out for powdered samples pressed into a sample holder. The patterns corrected for background scattering and normalized to the integrated intensity were analyzed using the WAXSFIT software [41,42]. During analysis, a theoretical curve composed from crystalline, meso-, and amorphous peaks approximating the experimental curve was constructed. The content of the ordered phases was calculated as the ratio of the integral intensity under crystalline and mesophase peaks to the total integral intensity scattered by the sample in the whole recorded range. For samples containing the β modification, the Kβ parameter characterizing the β-form content was calculated [43]. 

For SAXS investigations, a MBraun camera mounted directly on the top of the tube shield of a stabilized Philips PW 1830 X-ray generator was applied. In measurements, the CuKα radiation was used. Scattered radiation was registered by means of a MBraun linear position-sensitive detector (PSD 50) in the range of 0.02 ≤ s ≤ 0.8 (nm^−1^) (s = 2 sinθ/λ, 2θ the scattering angle and λ the X-ray wavelength) by the acquisition time of 1200 s. The measurements were performed for fibrous samples in the direction parallel to the fiber axis. Analysis of the SAXS data was conducted using the normalized one-dimensional correlation function [44]:(1)$$\gamma \left(r\right)=\frac{\int_{0}^{\infty }I\left(s\right)s^{2}\cos(2\pi rs)ds}{\int_{0}^{\infty }I\left(s\right)s^{2}ds}$$
In Equation (1), I(s) is scattering intensity and r represents the distance in the real space. Prior to integration, the data were extrapolated to zero and high angles, according to the procedure described in previous papers [45,46]. On the basis of the correlation function, the structural parameters, long period (L), thicknesses of crystalline (lC), and amorphous (lA) layers, were determined [44].

## 3. Results

### 3.1. Functionalized Reduced Graphene Oxide

#### 3.1.1. Morphology

Figure 2 shows the SEM images of graphite, GO, rGO, and rGO-CP. Natural graphite has a compact structure and is formed from flakes consisting of an indefinite number of closely adhering graphite layers (Figure 2a). The external dimensions of the flakes vary between few to over a dozen μm. As a result of oxidation, the compact structure of graphite becomes looser. The GO forms irregular agglomerates constituted from loosely connected rippled and buckled flakes, whose external dimensions vary between 75 and 220 μm (Figure 2b). As a result of micro-explosions occurring during reduction, the agglomerates of GO disintegrate and fine aggregates of rGO are formed. During thermal treatment, the oxygen-containing groups of GO rapidly decompose to produce gaseous products. The pressure generated from the evolving gases, mostly H_2_O and CO_2_, cause the rapid expansion of GO sheets. As a result, the aggregates of rGO reveal vermiculite morphology formed from ultra-thin wrinkled nanosheets (Figure 2c). The thickness of sheets is several times greater than the thickness of individual graphene layer. The aggregates exhibit greatly porous structure with the system of interconnected pores spreading from one side of the stack to another. 

After functionalization, some additional particles connected with the surface of rGO aggregates are visible (Figure 2d). The particles have irregular shapes and their dimensions do not exceed 1 μm.

#### 3.1.2. Chemical Structure

The changes observed in the morphology of graphene materials are strictly connected with the transformation of the chemical structure. Table 1 presents the content of C and O atoms after the subsequent stages of the treatment, determined on the basis of the EDS analysis. During oxidation, oxygenated functionalities are introduced into the graphite structure. The content of oxygen atoms exceeds 45% and the C/O ratio reaches 1.5, the typical value commonly observed in graphene oxide products [47]. Then, after reduction, the essential part of oxygen is removed. The content of the oxygen decreases and the atomic ratio C/O becomes greater than 4.

After functionalization, irregular particles containing calcium atoms are attached to the surface of rGO (Figure 3). The average atomic percentage of calcium atoms measured in different places amounts to several percent.

Figure 4 presents the FTIR spectra measured for graphite, GO, rGO, and rGO-CP.

No characteristic bands can be observed on the spectrum of pristine graphite. On the spectrum of GO in the range between 3800 and 2200 cm^−1^ overlapping bands corresponding to hydroxyl groups and adsorbed water molecules are visible. The band at 3427 cm^−1^ visible in this range corresponds to the stretching vibration of the hydroxyl groups placed on the surface of GO. The next band at 1725 cm^−1^ is related to stretching of carbonyl groups C=O of the carboxyl groups. The subjacent band at 1631 cm^−1^ is associated with the skeletal vibration of unoxidized graphitic domains. The band at 1393 cm^−1^ is assigned to deformation of the hydroxyl groups and bending vibration of interlayer water. The next two bands at 1231 and 1061 cm^−1^ correspond to CO (epoxy) and the CO (alkoxy) stretching vibrations [48].

On the spectrum of rGO, the number of visible bands has significantly decreased. In the range between 3800 and 2200 cm^−1^, the line becomes flat and the hydroxyl bands observed in this range are not visible after reduction. The band at 1725 cm^−1^ assigned to the COOH groups shows lower intensity. The next band, at 1600 cm^−1^, attributed to C=C stretching of aromatic bonds is relatively strong. The third band at 1275 cm^−1^ exhibits similar high intensity. The analysis of the FTIR spectra for rGO shows that after reduction the bands corresponding to hydroxyl groups disappear, while the intensity of the bands assigned to carboxyl groups decreases. It means that the hydroxyl groups occurring in the GO are eliminated and the carboxylic groups are partially removed. The intensification of the C=C band shows that after reduction the prominent sp2 graphitic domains are generated.

On the spectrum measured for rGO-CP, some additional bands are observed. According to the literature, the sharp band at 3640 cm^−1^ is related to the hydroxyl groups of unreacted calcium hydroxide [49]. Two strong bands at 2939 and 2861 cm^−1^ correspond to the stretching vibrations of carbon–hydrogen bonds in the aliphatic chains of pimelic acid. The band at 1706 cm^−1^ is assigned to the stretching vibration of the carbonyl group C=O in pimelic acid. The band at 1595 cm^−1^ is attributed to the symmetric and asymmetric stretching vibration of the carboxyl group in the metal salts. The next bands at 1435 and 1272 cm^−1^ are related to the oscillation vibrations of hydroxyl group and stretching of carbon–oxygen bond. The intensive band at 937 cm^−1^ is assigned to the bending vibrations of the plane of oxygen–hydrogen bond. The presence of the characteristic band in the 1560–1600 cm^−1^ range indicates that during the functionalization calcium pimelate is formed. The relatively large width of the band resulting from overlapping of two adjacent bands suggests that during reaction links involving calcium atoms connecting both carboxylic groups inside the calcium pimelate structure as well as groups from pimelic acid and groups occurring on the rGO layers are formed. 

#### 3.1.3. Nucleating Ability

Figure 5 presents the DSC curves registered during the non-isothermal crystallization of polypropylene without and with addition of different amounts of rGO-CP. On the curves, strong crystallization peaks of polypropylene is observed. For pure polypropylene, the maximum of the peak occurs at 115.1 °C (Table 2). For the polypropylene containing rGO-CP, the position of the crystallization peak is shifted to higher temperatures. The highest shift of 14 K is observed at medium additive concentration. The change of the peak position is connected with the changing nucleation temperature. In all samples with rGO-CP, crystallization starts at lower supercooling at the temperature which is significantly higher in comparison to the nucleation temperature of pure polypropylene.

The crystallization peak of the samples containing rGO-CP is sharper and its half-width is considerably smaller. The smallest half-width of the crystallization peak is observed for the highest additive concentration. The addition of rGO-CP results in the decrease of crystallization enthalpy. Again, the lowest value was registered at the highest additive concentration.

### 3.2. Structure of Polypropylene Fiber

#### 3.2.1. Differential Scanning Calorimetry (DSC)

Figure 6 shows the series of DSC curves registered for the polypropylene fibers extruded without and with the addition of rGO-CP taken at various take up velocities.

On the curves registered for the gravity spun fibers without the additive, a strong endothermic peak is observed at 165.6 °C. The peak is assigned to the melting of the crystalline α-iPP. For the fibers extruded with the addition of rGO-CP, the melting peak of α-iPP is minimally shifted towards higher temperatures. Except of the strong endothermic peak of the α-iPP in lower temperatures, two less intensive endothermic peaks attributed to the β-iPP are observed on the curves, approximately at 141 and 151 °C (Table 3). The peaks reflect two kinds of recrystallization occurring by heating of the samples containing the β-iPP. The first peak corresponds to ββ-recrystallization within the β-iPP, which results in the perfection of the β-iPP crystals. The second peak is related to βα-recrystallization, which involves the transition of the β-iPP into more thermodynamically stable α-iPP [19].

Two additional endothermic peaks are visible for all samples containing rGO-CP. The position of the peaks and enthalpies of melting are related to additive concentration. With the increase of the additive concentration the position of peaks is shifted slightly towards higher temperatures. Simultaneously, the enthalpy of melting of the β-iPP as well as the total degree of crystallinity increase considerably. 

On the curves registered for all fibers taken at 50 m/min, the intense endothermic peak related to the melting of the α-iPP occurs. In this case, despite the addition of rGO-CP, the endothermic peaks of the β-iPP observed for the gravity spun fibers are not visible. With the increase of the additive concentration, the endothermic peak moves slightly towards higher temperatures (Table 4). 

The peak of the α-iPP melting is accompanied by an extensive exothermic peak connected with the recrystallization of the mesomorphic phase. The peak is located between 60 and 150 °C with the maximum at 102 °C. For the fibers extruded without additive, the exothermic peak has a relatively high intensity. Similar high intensity was recorded for the peak registered for the fibers containing the lowest additive concentration (0.1 wt.%). Consequently, for these both fibers the measured enthalpy of mesophase recrystallisation achieves high value. With higher additive concentrations the intensity of the exothermic peak decreases. Simultaneously, enthalpy associated with the mesophase transition decreases and its value for the highest additive concentration (1 wt.%) is low, almost four times lower compared to the fibers extruded without additives. Regardless of the rGO-CP addition, for all the fibers taken at 50 m/min the degree of crystallinity achieves the same value. 

The curves measured for the fibers taken at 200 m/min are similar to the curves measured for the fibers taken at 50 m/min. The strong endothermic peak associated with melting of α-iPP is accompanied by the diffuse exothermic peak associated with the mesophase transition. The lowest mesophase recrystallisation enthalpy occurs in fibers extruded without additives. With an additive, the enthalpy slightly increases and achieves the highest value when the additive concentration is the highest (1 wt.%). With the growth of the rGO-CP concentration the degree of crystallinity decreases minimally.

On the curves recorded for all fibers taken at 800 m/min, only one intense endothermic peak related to the melting of α-iPP is visible. Other endothermic peaks associated with β-iPP observed in gravity spun fibers as well as exothermic peaks associated with mesophase recrystallisation occurring for fibers taken at low and medium velocities are not visible in this case. In comparison to the curves registered for lower velocities, the endothermic peak is much sharper and narrower. The peak location is similar to the position of the peak measured at medium velocity (Table 5). The degree of crystallinity slightly fluctuates with the addition of the rGO-CP.

#### 3.2.2. Wide Angle X-Ray Scattering (WAXS)

Figure 7 shows the series of WAXS patterns of pure polypropylene fibers and fibers with rGO-CP additive, taken at various take up velocities.

The WAXS pattern of gravity spun fibers extruded without additives reveals several crystalline peaks characteristic for the α-iPP. For the fibers doped with rGO-CP, beside the peaks attributed to the α-iPP at 16.1°, a distinct diffraction peak assigned to the (300) crystal plane of the β-iPP is observed. The intensity of this peak is the highest when the rGO-CP content is the lowest (0.1 wt.%). With the increase of the rGO-CP concentration the peak intensity slightly decreases. 

In the case of the fibers taken at 50 m/min, two broad peaks attributed to mesophase are observed for pure polypropylene at 15° and 21°. The mesophase peaks have high intensity and overlap weaker crystalline peaks. In this situation, the crystalline peaks are not visible at all. A similar pattern with two broad mesophase peaks is observed for the fibers extruded with rGO-CP at the lowest additive concentration. For the medium rGO-CP content (0.5 wt.%), the mesophase peaks become weaker and are overlapped by stronger crystalline peaks assigned to the α-iPP. For the highest rGO-CP content (1 wt.%), the mesophase peaks are no longer visible. The pattern shows strong crystalline peaks of the α-iPP with an additional weak peak of the β-iPP. 

For the fibers taken at 200 m/min, the strong crystalline peaks of the α-iPP and the weak mesophase peaks overlap. Regardless of the addition of the rGO-CP, diffraction patterns for all the fibers are similar.

For the fibers taken at the highest velocity of 800 m/min, WAXS patterns reveal only sharp and strong crystalline peaks of the α-iPP. The pattern is the same for all the investigated fibers extruded without and with additives.

Figure 8 presents examples of an analysis procedure applied for the WAXS patterns. The analysis of the gravity spun fibers doped with rGO-CP includes several crystalline peaks characteristic for the α-iPP and the β-iPP (Figure 8a). The analysis of the fibers containing mesophase covered the crystalline of the α-iPP and mesophase peaks (Figure 8b). 

Based on the analysis performed, the content of ordered phases and the Kβ value were calculated. The parameters are presented in Table 6.

For the gravity spun fibers extruded without additives, the crystalline structure containing crystals of the α-iPP is formed. For these fibers, the crystallinity index achieves a high value of 47.5%. The crystallinity index is higher for the fibers extruded with addition of the rGO-CP and increases with the increase of the additive concentration. Adding rGO-CP results in a significant amount of the β-iPP formed in the fiber next to the α-iPP. The Kβ value, characterizing the content of the β-iPP, reaches the highest level at the lowest additive concentration (0.1 wt%). For medium (0.5 wt%) and high (1 wt%) rGO-CP concentrations, the Kβ value is considerably lower. 

At the lowest velocity (50 m/min), the structure with high content of mesophase is formed in the fibers extruded without additives. A similar structure is formed inside the fibers containing a small amount of the rGO-CP. At higher additive concentrations, a more ordered structure is formed instead of the mesophase. This change does not involve a change in the total content of ordered phases and, regardless of the additive concentration, the content of the ordered phases remains at a constant level. With the highest rGO-CP concentration, the small amount of the β-iPP is observed. 

At medium velocity (200 m/min), the crystalline structure with the addition of a small content of mesophase is formed regardless the addition of the rGO-CP. The content of the ordered phases is minimal in comparison to the fibers taken at lower velocity (50 m/min) and decreases minimally with the increase of the additive concentration. 

At the highest velocity (800 m/min), the high crystalline structure without mesophase is formed. The crystallinity achieves a high value—higher than the content of ordered phases determined for the lower velocities. At low and medium additive concentrations (0.1 and 0.5 wt.%), the crystallinity index is practically the same as for the fibers extruded without additives. At a high concentration the crystallinity index is lower.

#### 3.2.3. Small Angle X-Ray Scattering (SAXS)

Figure 9 shows the series of SAXS diffraction spectra for the investigated fibers. For all curves a distinct maximum of long the period related to the lamellar structure of polypropylene is observed. The angular position of the maximum changes slightly, indicating changes in the long period of the structure.

Figure 10 presents a typical picture of one-dimensional correlation function derived from SAXS curves according to equation (1) for the fibers taken at 200 m/min. 

Parameters determined on the basis of the correlation functions are presented in Table 7. In case of the gravity spun fibers, the long period measured for the fibers containing rGO-CP is considerably greater than for the fibers extruded without additives. For these fibers an increase in both, the thickness of the crystalline lamellae and the thickness of the amorphous layers, is also observed. For the fibers taken at 50 m/min, a smaller increase of measured parameters is detected. For the fibers taken at 200 m/min, regardless of the rGO-CP content, the values of the long period as well as the crystalline and amorphous thickness are practically constant. For the fibers taken at 800 m/min, a slight decrease in the value of both, the long period and the thickness of the crystal lamellae, is observed with the increase of the rGO-CP content.

## 4. Discussion

Oxidation of graphite powder followed by thermal reduction allows to obtain reduced graphene. The compound possesses the sufficient amount of carboxylic groups capable of functionalization. During the functionalization, calcium pimelate (CP) is formed by reaction of pimelic acid with calcium hydroxide. Simultaneously, formation of intermolecular bonds between the carboxylic groups of reduced graphene oxide and pimelate acid with the participation of calcium atoms is initiated. Due to these bonds, particles of calcium pimelate are attached to the surface of reduced graphene oxide. Formation of permanent bonds was confirmed in previous investigations [35].

By adding the rGO functionalized with calcium pimelate into polypropylene melt, the non-isothermal crystallization occurs at higher temperature (Figure 5). The significant shift of the crystallization temperature of about 12 K is observed already at low additive concentration. At medium concentration, the shift is 2 K higher, while for the highest concentration it is again 3.5 K lower. Generally, the increase of crystallization temperature equals approximately 12 K, which corresponds well to the increment of the crystallization temperature observed for organic pigments [50], metal salts [51], and other efficient nucleating agents [52].

During fiber formation through cooling the extruded threads the polypropylene crystallization occurs. Crystallization proceeds in non-isothermal conditions and under flow field. It is known that flow conditions strongly affect polypropylene crystallization and its influence is reflected in the fiber structure.

Gravity spun fibers are taken without additional drawing and polypropylene crystallization occurs under the minimal influence of molecular orientation. In these conditions, nucleation proceeds isotopically, similar to crystallization occurring in the quiescent state. Addition of the rGO-CP to polypropylene melt results in the formation of crystalline structure with higher crystallinity. The rGO-CP acts as a typical nucleating agent, leading to formation of interfacial layer with reduced chain mobility. With the introduction of additives the nucleation density increases and the structure with higher crystallinity is formed. The crystallinity of fibers grows with the increase of the additive concentration. 

Due to its specific surface geometry, calcium pimelate is well known as efficient nucleating agent promoting growth of the β-iPP [30]. Previous investigations revealed that calcium pimelate connected with the surface of rGO initiates the growth of the β-form crystals and promotes formation of structure with a considerable amount of the β-iPP [35]. This ability of the rGO-CP is clearly manifested in gravity spun fibers. In the fibers extruded without additives, the supramolecular structure formed consists only of the α-iPP crystals. On the contrary, in fibers containing rGO-CP, a considerable amount of the β-iPP appears in addition to the α-iPP crystals. Unexpectedly, the highest content of the β-iPP is observed at the lowest rGO-CP concentration. For medium and high rGO-CP concentrations, the content of the β-iPP is much lower. Lower content of the β-iPP at higher additive concentration may be caused by inhomogeneous distribution of calcium pimelate on the surface of rGO aggregates and/or the poor dispersion of the rGO-CP inside the polypropylene matrix. The reason for the decrease of the β-iPP content for higher additive concentration is not clear and further examinations are required. 

For the next three series of fibers taken at 50, 200, and 800 m/min, polypropylene crystallization is more affected by molecular orientation. The orientation of polypropylene chains influences both, the nucleation mechanism and overall crystallization kinetics. 

For the lowest take-up velocity (50 m/min), the influence of the orientation is minimal. In these conditions, in fibers extruded without additives, the structure with a high content of mesophase is formed. The presence of mesophase is manifested both in WAXS patterns and on DSC curves (Figure 6b and Figure 7b). The mesophase was repeatedly obtained in the polypropylene fibers taken at low and moderate velocities. In particular cases, the formation window of take-up velocities resulting in mesophase formation was determined [53]. 

The structure with dominating mesophase is also obtained in fibers containing small amount of the rGO-CP (0.1 wt.%). For medium additive concentration (0.5 wt.%), the mesophase content is much smaller and the structure with a bigger content of crystalline phase is formed. For high additive concentration (1 wt.%), the mesophase disappears and only the crystalline structure is observed. The changes registered in the fiber structure indicate that rGO-CP has meaningful influence on polypropylene crystallization in the fibers taken at low take-up velocity. At medium concentration, the content of the crystalline phase significantly increases as a result of nucleating activity of the additive, whereas at high concentration even a small amount of the β-iPP is obtained. 

For medium take-up velocity (200 m/min), the impact of the molecular orientation on the polypropylene crystallization is significantly greater. Crystallization in these conditions results in the decreasing content of the mesophase. Regardless of the additive addition and the additive concentration, the structure with dominating crystalline phase is formed in all fibers.

For the highest velocity (800 m/min), polypropylene crystallization is governed by high orientation. Due to orientation, the segments of polypropylene chains are subject to straightening. Bundles of straightened segments form active row nuclei which grow quickly forming lamellar crystals of the α-form. The row nuclei are formed regardless of the presence of the additive. The particles of the rGO-CP do not take participation in the nucleation and do not influence on polypropylene crystallization. In all the fibers, crystallization proceeds similarly and a similar structure characterized by the same parameters is obtained. 

## 5. Conclusions

An effective β-nucleating agent can be obtained by functionalization of the rGO with pimelic acid and calcium hydroxide. Calcium pimelate attached to surface of the rGO increases the crystallization temperature in the range comparable with other effective nucleating agents and promotes selective growth of the β-iPP. Used as the additive, rGO-CP reveals its nucleating ability in gravity spun fibers during the formation of polypropylene fibers. In the presence of rGO-CP, a considerable amount of the β-iPP is obtained. In this case, the amount of the β-iPP depends on the additive concentration. In fibers taken at higher velocities, molecular orientation has an increasing impact on the formation of the fiber structure. With the increase of the take-up velocity, the influence of orientation increases and the impact of rGO-CP on the fiber structure gradually decreases. At the highest investigated velocity, crystallization is governed by high orientation and the nucleating effect of the additive loses its importance. 

## Figures and Tables

**Figure 1 polymers-12-00910-f001:**
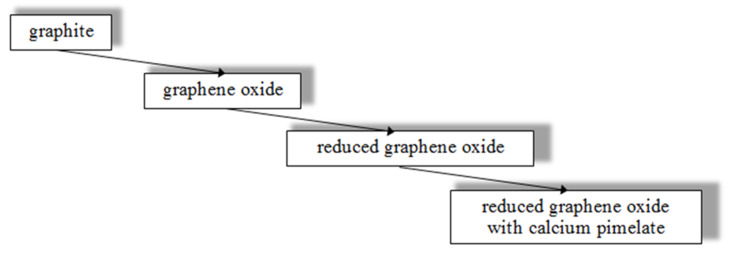
Scheme for preparation of functionalised reduced graphene oxide.

**Figure 2 polymers-12-00910-f002:**
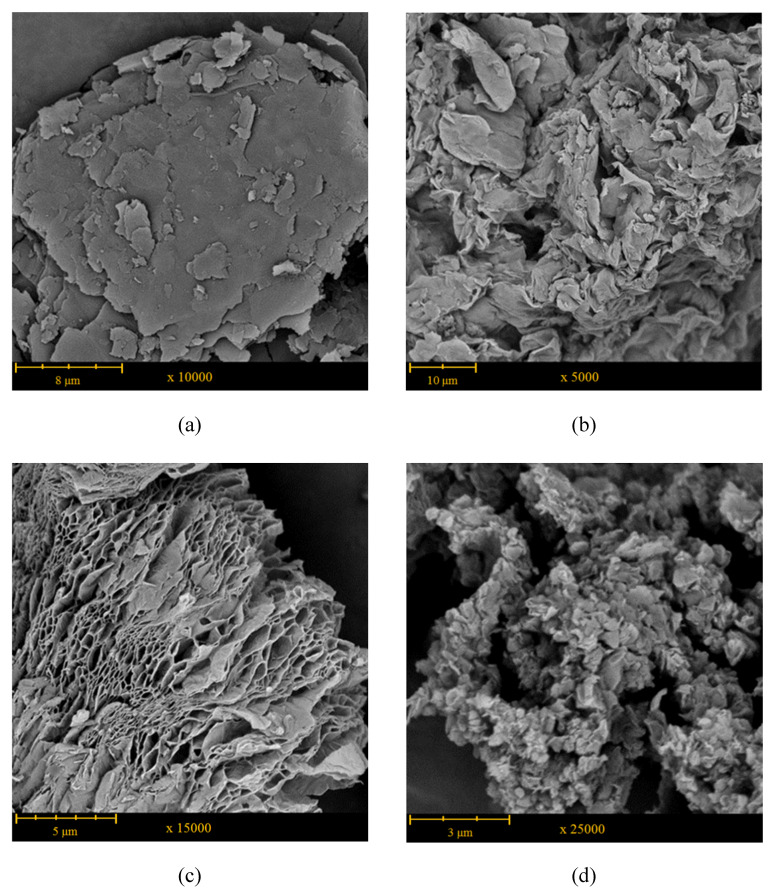
SEM micrographs of: (**a**) graphite; (**b**) graphene oxide (GO); (**c**) reduced graphene oxide (rGO); (**d**) functionalized reduced graphene oxide (rGO-CP).

**Figure 3 polymers-12-00910-f003:**
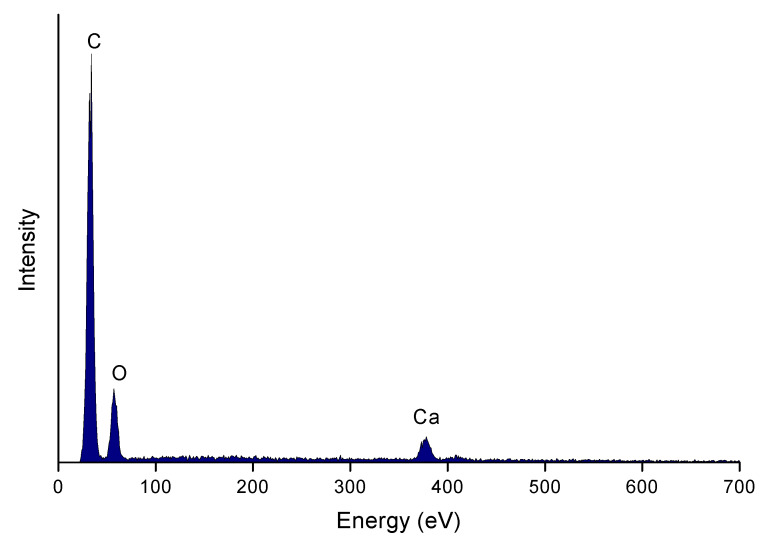
EDS spectra for functionalized reduced graphene oxide (rGO-CP).

**Figure 4 polymers-12-00910-f004:**
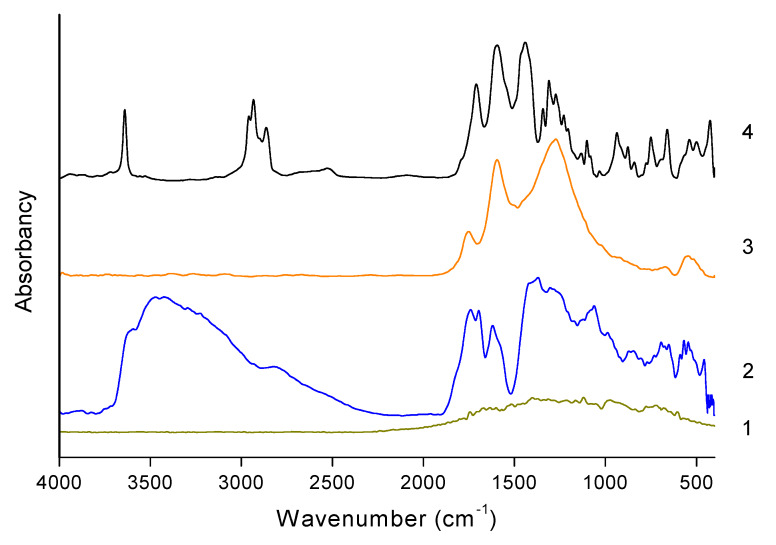
FTIR spectra of graphite (1), graphene oxide GO (2), reduced graphene oxide rGO (3), and reduced graphene oxide functionalized with calcium pimelate rGO-CP (4).

**Figure 5 polymers-12-00910-f005:**
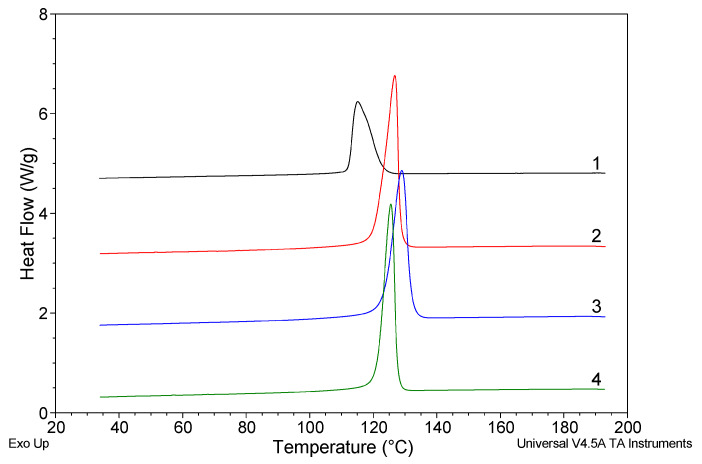
DSC curves during cooling of polypropylene without rGO-CP—black (1); polypropylene + rGO-CP (0.1%)—red (2); polypropylene + rGO-CP (0.5%)—blue (3); polypropylene + rGO-CP (1.0%)—green (4).

**Figure 6 polymers-12-00910-f006:**
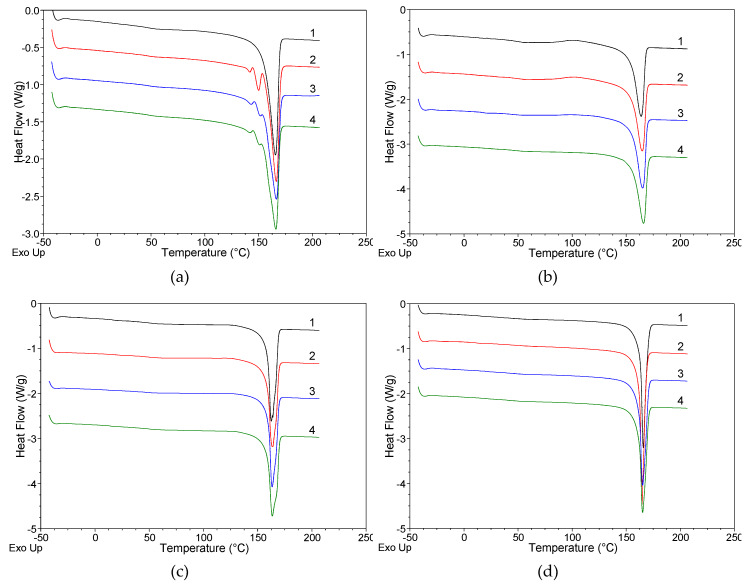
DSC curves for polypropylene fibers: (**a**) gravity spun; fibers taken at (**b**) 50 m/min; (**c**) 200 m/min; (**d**) 800 m/min. In the drawings, black curves (1) refer to fibers of pure polypropylene, while red (2), blue (3), and green (4) curves refer to fibers of iPP with rGO-CP in the amount of 0.1, 0.5, and 1.0 wt.%, respectively.

**Figure 7 polymers-12-00910-f007:**
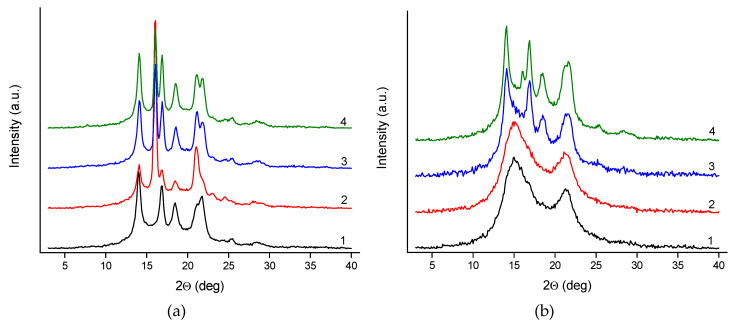
Wide-angle X-ray scattering (WAXS) patterns of polypropylene fibers: (**a**) gravity spun; fibers taken at (**b**) 50 m/min; (**c**) 200 m/min; (**d**) 800 m/min. In the drawings, black curves (1) refer to fibers of pure polypropylene, while red (2), blue (3), and green (4) curves refer to fibers of iPP with rGO-CP in the amount of 0.1, 0.5, and 1.0 wt.%, respectively. (The curves were shifted along the intensity axis for clarity).

**Figure 8 polymers-12-00910-f008:**
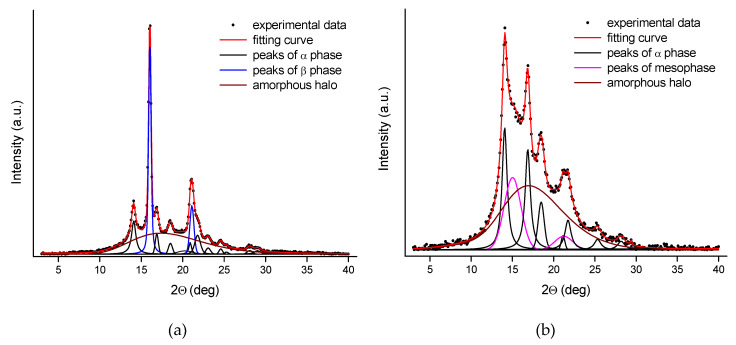
Analysis of WAXS patterns; (**a**) gravity spun fibers with addition of rGO-CP (0.1 wt.%); (**b**) fibers taken at 200 m/min.

**Figure 9 polymers-12-00910-f009:**
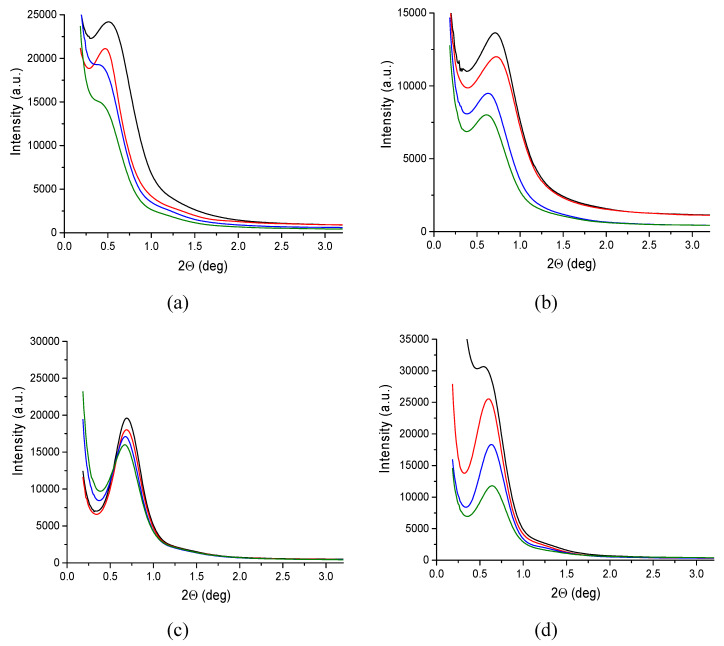
Small-angle X-ray scattering (SAXS) patterns of polypropylene fibers: (**a**) gravity spun; fibers taken at (**b**) 50 m/min; (**c**) 200 m/min; (**d**) 800 m/min. In the drawings, black curves refer to fibers of pure polypropylene, while red, blue, and green curves refer to fibers of iPP with rGO-CP in the amount of 0.1, 0.5, and 1.0 wt.%, respectively.

**Figure 10 polymers-12-00910-f010:**
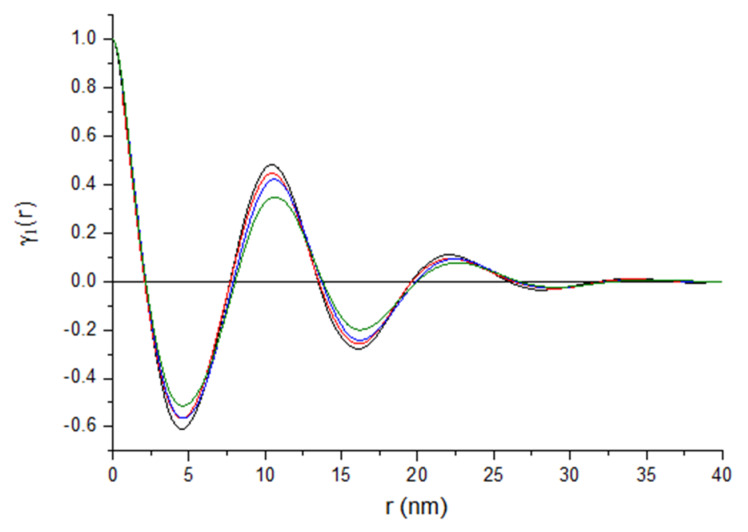
One-dimensional correlation functions for fibers taken at 200 m/min.

**Table 1 polymers-12-00910-t001:** Atomic percentage of atoms in graphene-based materials.

	Element Content		
	C [%]	O [%]	Ca [%]
graphite	100	-	-
GO	53.5	46.5	-
rGO	75.8	24.2	-
rGO-CP	74.3	23.2	2.5

**Table 2 polymers-12-00910-t002:** Parameters of polypropylene crystallization.

Sample	Temperature of Crystallization	Temperature of Nucleation	Half-width of the Crystallization Peak	Enthalpy of Crystallization
	*T_c_* [°C]	*T_c ON SET_* [°C]	Δ*T_c(0,5H)_* [°C]	Δ*H_c_* [J/g]
iPP 100%	115.1	122.8	6.4	114.90
iPP + rGO-CP (0.1 wt.%)	126.8	128.5	4.4	108.04
iPP + rGO-CP (0.5 wt.%)	129.0	132.0	5.1	107.89
iPP + rGO-CP (1.0 wt.%)	125.5	127.5	3.7	98.63

**Table 3 polymers-12-00910-t003:** Parameters determined on the basis of DSC measurements for gravity spun fibers.

Sample	Temp. Of Melting β-iPP		Enthalpy of Melting β-iPP	Temp. of Melting α-iPP	Enthalpy of Melting α-iPP	Total Degree of Crystall.
	*T_m1_**_β_* [°C]	*T_m2_**_β_* [°C]	Δ*H_m_* *_β_* [J/g]	*T_m_**_α_* [°C]	Δ*H_m_* *_α_* [J/g]	*κ _DSC_* [%]
iPP 100%	––	––	––	165.6	97.54	47.1
iPP + rGO-CP (0.1 wt.%)	141.7	150.0	20.45	166.3	82.63	50.3
iPP + rGO-CP (0.5 wt.%)	141.9	150.7	21.88	167.2	82.59	51.2
iPP + rGO-CP (1.0 wt.%)	142.9	151.5	23.36	166.4	86.21	54.0

**Table 4 polymers-12-00910-t004:** Parameters determined on the basis of DSC measurements for fibers taken at 50 and 200 m/min.

Take-up Velocity [m/min]	Sample	Temp. of Recrystall. Mesophase	Enthalpy of Recrystall. Mesophase	Temp. of Melting	Enthalpy of Melting	Total Degree of Crystall.
		*T_m__m_* [°C]	Δ*H_m__m_* [J/g]	*T_m_* [°C]	Δ*H_m_* [J/g]	*κ _DSC_* [%]
	iPP 100%	102.3	19.26	163.8	96.76	46.7
	iPP + rGO-CP (0.1 wt.%)	103.4	18.84	164.9	96.61	46.7
50	iPP + rGO-CP (0.5 wt.%)	102.3	12.31	165.2	96.08	46.8
	iPP + rGO-CP (1.0 wt.%)	102.4	5.13	166.0	96.14	46.9
	iPP 100%	124.2	9.64	162.5	101.77	49.2
	iPP + rGO-CP (0.1 wt.%)	123.0	11.11	163.3	100.60	48.7
200	iPP + rGO-CP (0.5 wt.%)	125.5	11.80	163.3	96.08	46.6
	iPP + rGO-CP (1.0 wt.%)	126.7	12.49	163.4	94.91	46.3

**Table 5 polymers-12-00910-t005:** Parameters determined based on DSC measurements for fibers taken at 800 m/min.

Sample of Fibers	Temp. of Melting	Enthalpy of Melting	Total Degree of Crystall.
	*T_m_* [°C]	Δ*H_m_* [J/g]	*κ _DSC_* [%]
iPP 100%	162.5	101.77	49.2
iPP + rGO-CP (0.1 wt.%)	163.3	100.60	48.6
iPP + rGO-CP (0.5 wt.%)	163.3	99.41	48.0
iPP + rGO-CP (1.0 wt.%)	163.4	99.97	48.3

**Table 6 polymers-12-00910-t006:** Crystalline structure parameters obtained by means of the WAXS method.

Take-Up Velocity [m/min]	rGO-CP Content [wt.%]	Content of Ordered Phase κ_WAXS_ [%]	K_β_ [ - ]
Gravityspun	0	47.5	-
	0.1	49.7	0.67
	0.5	50.9	0.28
	1	54.3	0.31
50	0	47.0	-
	0.1	47.1	-
	0.5	47.3	-
	1	47.4	0.09
200	0	49.5	-
	0.1	49.1	-
	0.5	47.1	-
	1	46.8	-
800	0	50.6	-
	0.1	50.4	-
	0.5	50.3	-
	1	50.2	-

**Table 7 polymers-12-00910-t007:** Lamellar structure parameters of investigated fibers obtained by means of SAXS method.

Take-up Velocity [m/min.]	rGO-CP Content [%]	L [nm]	l_C_ [nm]	l_A_ [nm]
gravity spun	0	11.6	2.9	8.7
	0.1	13.1	3.4	9.7
	0.5	13.4	3.6	9.8
	1	13.3	3.5	9.8
50	0	9.2	2.6	6.6
	0.1	9.1	2.7	6.4
	0.5	10.4	2.8	7.6
	1	10.7	2.9	7.8
200	0	10.4	2.9	7.5
	0.1	10.4	2.9	7.5
	0.5	10.6	3.0	7.6
	1	10.7	3.0	7.7
800	0	11.8	4.0	7.8
	0.1	11.7	3.4	8.3
	0.5	11.5	3.1	8.4
	1	11.1	3.0	8.1

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
