# Peer review of "Supramolecular Structure of Polypropylene Fibers Extruded with Addition of Functionalized Reduced Graphene Oxide"

_polymers, 2020, doi:10.3390/polym12040910_

Round 1

Reviewer 1 Report

In this manuscript, the authors present the changes of supramolecular structure of the PP fibres extruded with b-nucleating agent – reduced grapheme oxide – during non-isothermal crystallization at their preparation. This structure is compared with the supramolecular structure of the pure PP fibre. The results confirm the b-nucleating ability of reduced graphene oxide for the gravity spun fibres and fibres taken at lower take-up velocity.

The manuscript is very interesting but needs the minor corrections. After minor revision I would highly recommend the acceptance of this work in Polymers.

  1. points 171 and 301, Table 3 – to explain the calculation of “Total degree of crystal. KDSC
  2. points 581, References 37 – correct name of journal
  3. to improve the quality of lines and axis title of all figures.

Author Response

  1. points 171 and 301, Table 3 – to explain the calculation of “Total degree of crystal. KDSC

Total degree of crystallinity was calculated as a sum of contents of two polypropylene modifications: α-iPP and β-iPP determined on the basis of DSC curves.  The term was used to underline the presence of both modifications and their input in the samples crystallinity. To avoid confusion an additional sentence with explanation in the experimental section was added.

  1. points 581, References 37 – correct name of journal

The font for name of journal was changed.

  1. to improve the quality of lines and axis title of all figures.

According to reviewer comment the quality of all figures was improved.

Reviewer 2 Report

  1. The authors should point out the main innovation points of their work, not just conclude what they have done.
  2. The section of "Materials and Methods" should be modified carefully. Devided into sub-headings including "2.1 Materials", "2.2 Preparation of ...", and "2.3 Characterization" will be much better for reading.
  3. I have to say that most of the numbers and titles of the coordinate axis are not clear.
  4. Put the discussion section along with the corresponding images is better for understanding.

Author Response

  1. The authors should point out the main innovation points of their work, not just conclude what they have done.

During investigations the functionalized reduced graphene oxide with nucleating ability towards the β-iPP was obtained.  To our knowledge the β-nucleating agent for polypropylene based on reduced graphene oxide functionalised with calcium pimelate was obtained for the first time. The nucleating ability of the compound in polypropylene crystallisation in quiescent state was examined in our laboratory. The results of this examinations were presented in previous paper.  At the next stage of research the functionalised reduced graphene oxide was used as an additive by melt spinning of the polypropylene fibres. The influence of the additive on the structure of fibres extruded at different parameters was analysed. The results are presented in the paper.  Such examinations were not performed so far and the paper is the first publication on this topic. To underline the innovation point of the research additional sentences in the introduction were introduced.

  1. The section of "Materials and Methods" should be modified carefully. Devided into sub-headings including "2.1 Materials", "2.2 Preparation of ...", and "2.3 Characterization" will be much better for reading.

According to reviewer suggestion the section "Materials and Methods"  was divided into sub-headings.

  1. I have to say that most of the numbers and titles of the coordinate axis are not clear.

According to reviewer comment the quality of all figures was improved.

  1. Put the discussion section along with the corresponding images is better for understanding.

In the research complementary methods were used. The results refers to different aspects of supramolecular structure of polypropylene fibres. The results obtained by various methods are compatible and support each other.  In authors opinion the discussion presented in the paper summarizes all obtained results and comments the results of examinations in the best way. To facilitate understanding some references to corresponding images were added.

Round 2

Reviewer 2 Report

it can be accepted now